# TEST-TIME TUNED LANGUAGE MODELS ENABLE END-TO-END DE NOVO MOLECULAR STRUCTURE GENERATION FROM MS/MS SPECTRA

**Laura Mismetti**
IBM Research, Zürich
ETH Zürich, Switzerland
laura.mismetti1@ibm.com

**Marvin Alberts**
IBM Research, Zürich
UZH, Zürich, Switzerland

**Andreas Krause**
ETH Zürich, Switzerland

**Mara Graziani**
IBM Research, Zürich

## ABSTRACT

Tandem Mass Spectrometry is a cornerstone technique for identifying unknown small molecules in fields such as metabolomics, natural product discovery and environmental analysis. However, certain aspects, such as the probabilistic fragmentation process and size of the chemical space, make structure elucidation from such spectra highly challenging, particularly when there is a shift between the deployment and training conditions. Current methods rely on database matching of previously observed spectra of known molecules and multi-step pipelines requiring intermediate fingerprint prediction or expensive fragment annotations. We introduce a novel end-to-end framework based on a transformer model that directly generates molecular structures from an input tandem mass spectrum and its corresponding molecular formula, thereby eliminating the need for manual annotations and intermediate steps, while leveraging transfer learning from simulated data. To further address the challenge of out-of-distribution spectra, we introduce a test-time tuning strategy that dynamically adapts the pre-trained model to novel experimental data. Our approach achieves a Top–1 accuracy of 3.16% on MassSpecGym benchmark and 12.88% on NPLIB1 datasets, considerably outperforming conventional fine-tuning. Baseline approaches are surpassed by 27% and 67% respectively. Even when the exact reference structure is not recovered, the generated candidates are chemically informative, exhibiting high structural plausibility as reflected by strong Tanimoto similarity to the ground truth. Notably, we observe a relative improvement in average Tanimoto similarity of 83% on NPLIB1 and 64% on MassSpecGym compared to state-of-the-art methods. Our framework combines simplicity with adaptability, generating accurate molecular candidates that offer valuable guidance for expert interpretation of unseen spectra.

## 1 INTRODUCTION

Deciphering the molecular structure of an unknown compound from spectroscopic data is one of the most challenging puzzles in analytical chemistry, where Nuclear Magnetic Resonance (NMR), Infrared (IR) and Tandem Mass (MS/MS) spectroscopy (Weatherly et al., 2005; Kapp & Schütz, 2007) provide complementary evidence to reconstruct the underlying structure. Classical workflows accelerate discovery via heuristic database matching against known references (Dührkop et al., 2019; Wang et al., 2020; Li et al., 2025; Dührkop, 2022), yet they struggle to scale with the combinatorial breadth of chemical diversity and the effectively unbounded chemical space. Artificial Intelligence (AI) is increasingly effective for de-novo generation in Chemistry (Schwaller et al., 2019; Born & Manica, 2023; Frieder et al., 2023), and has demonstrated potential for structure elucidation from NMR (Jonas, 2019; Sridharan et al., 2022; Alberts et al., 2023; Schilter et al., 2023; Hu et al., 2024; Devata et al., 2024; Alberts et al., 2025b), IR (Fine et al., 2020; Enders et al., 2021; Alberts et al., 2024a; 2025c; Wu et al., 2025), and multimodal approaches combining diverse spectra simultaneously (Priessner et al., 2024; Alberts et al., 2025a).

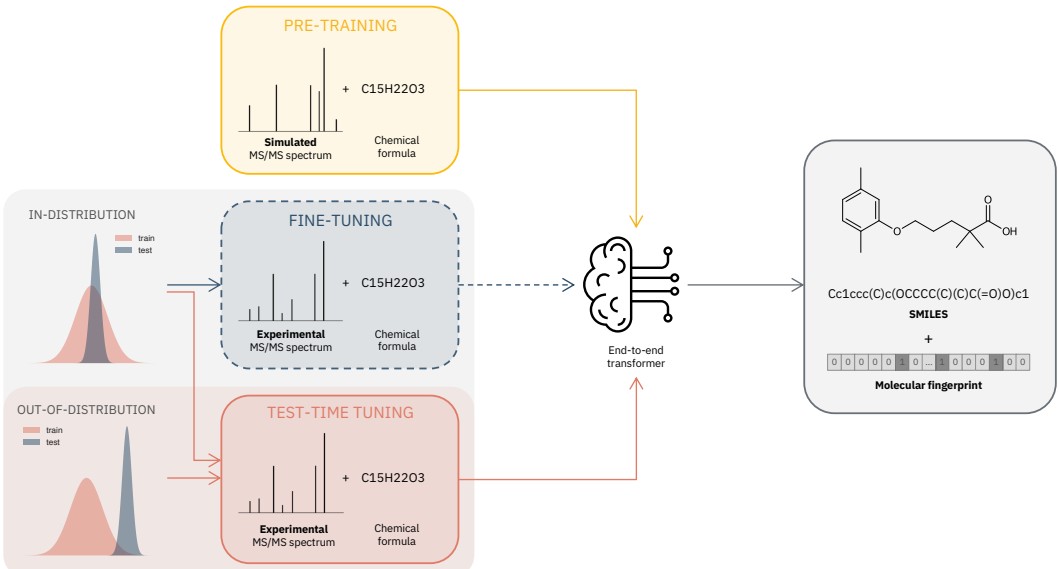

Figure 1: We propose a transformer encoder–decoder model that predicts SMILES and molecular fingerprint from a MS/MS spectrum and its associated chemical formula. The model is pre-trained on simulated spectra (Alberts et al., 2024b) and adapted via test-time tuning on experimental data from NPLIB1 (Dührkop et al., 2021) and MassSpecGym (Bushuiev et al., 2024). We compare standard fine-tuning to our proposed test-time tuning strategy, which selects the training samples that are the most informative about the target experimental spectra based on the prediction of molecular fingerprints, in in-distribution and out-of-distribution scenarios.

Existing methods to perform structure elucidation from MS/MS spectra either rely on expert-curated fragment annotations or learn spectral fingerprints, both of which limit the applicability to novel compounds (Huber et al., 2021a;b). Other approaches predict molecular structures either passing through molecular fingerprints (Goldman et al., 2024b; Dührkop, 2022) or directly from spectra (Litsa et al., 2023; Butler et al., 2023; Shrivastava et al., 2021; Wang et al., 2025). However, robust generalization remains a bottleneck. The MassSpecGym benchmark (Bohde et al., 2025) makes this complexity explicit: despite substantial methodological diversity, reported Top–1 accuracies remain low, underscoring the persistent challenge of reliable structure reconstruction from MS/MS alone. Among leading approaches, MSNovelist casts the problem as a sequence-to-sequence task that generates SMILES (Stravs et al., 2022), MADGEN first retrieves the molecular scaffold, which is then used as starting point for a generative model (Wang et al., 2025), whereas DiffMS first learns molecular fingerprints and then performs iterative diffusion-based reconstruction, yielding the strongest reported MassSpecGym Top–1 accuracy to date at 2.30% (Bohde et al., 2025). The main challenge of the MassSpecGym dataset lies in the domain shift between the training and test sets. Molecules present in the test set exhibit considerable differences from those found in the training set, an important challenge that standard fine-tuning leaves unaddressed (see Figure 3). This discrepancy matches realistic expectations of structure elucidation models, where the spectra used in real-world applications can differ substantially from the reference data used for training.

Transductive learning is a technique widely used in other fields to mitigate this issue (Gammerman et al., 1998; Farahani et al., 2021). It leverages unlabeled target-domain samples to adapt the model at inference time by selecting the most informative points from a candidate pool, typically the available training set, and training only on these selected samples. We evaluate whether this common domain adaptation strategy can successfully narrow the domain gap observed in Tandem Mass spectroscopy. Since each target experimental spectrum is available, and only its structure is unknown, it can be used as an unlabeled sample to adapt the model at test time (Sun et al., 2020) to be more robust to eventual shifts in the test distribution (Hübotter et al., 2025). Crucially, while test-time training has been applied in proteomics for MS/MS spectrum prediction (Ye et al., 2024), it has not been applied to de novo small-molecule structure generation, which is the task we target here.

We introduce a novel framework for structure elucidation from MS/MS spectra (see Figure 1) that, unlike existing approaches (Stravs et al., 2022; Bohde et al., 2025), eliminates the need for intermediate annotations or predicted fragments and reports consistent accuracy improvements across all benchmarks. To address variability across datasets, we explore two adaptation strategies: classical fine-tuning and test-time tuning. Our method builds on a transformer encoder–decoder architecture, pre-trained on a large corpus of simulated spectra (Alberts et al., 2024b; Dorigatti et al., 2025) and leverages predicted molecular fingerprints to improve structural consistency. The latter are key in the test-time tuning strategy to identify within the training data which samples are the most informative about the experimental target data and should therefore be used for tuning the model parameters. Compared to traditional fine-tuning, test-time tuning is particularly effective on the out-of-distribution spectra in MassSpecGym, and standard fine-tuning suffices when the train and test distributions are aligned. Even when predictions deviate from the reference structure, our generated candidates remain chemically informative, compared to those obtained from existing methods, providing valuable guidance to make an informed guess about the compound. These results demonstrate that our framework has the potential to substantially streamline structure elucidation routines from MS/MS spectra, facilitating its integration into high-throughput workflows where rapid and accurate identification of unknown compounds is essential.

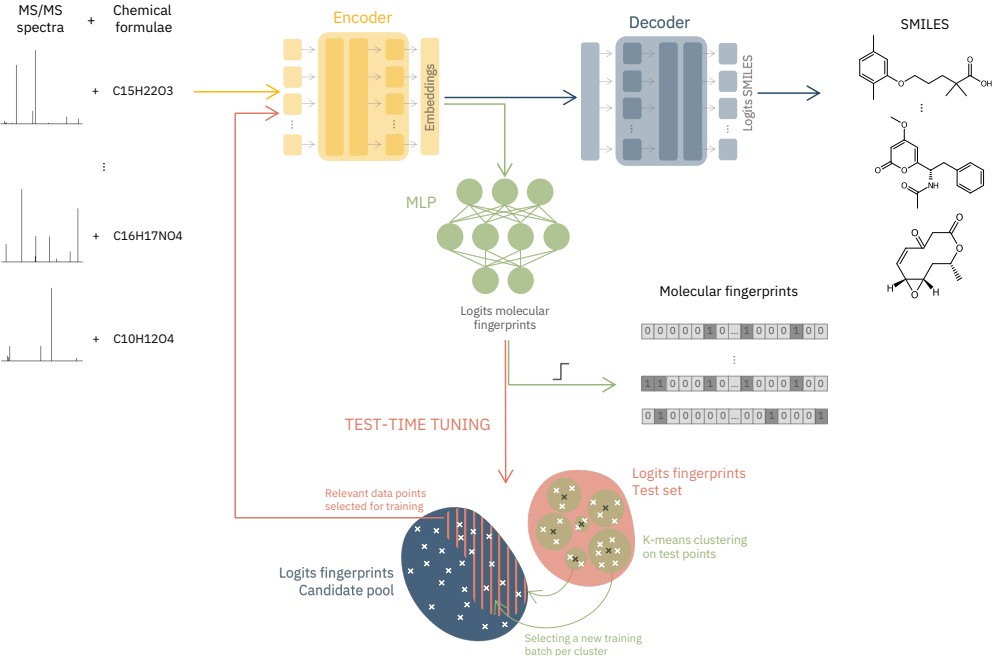

Figure 2: Schematic illustration of test-time tuning workflow: MS/MS spectrum and chemical formula are the input of the transformer encoder–decoder which predicts SMILES. The encoder generates embeddings used as input to a multilayer perceptron (MLP) trained to predict molecular fingerprints through an additional loss term. The logits produced by the MLP are the projection into a chemical feature space, and this representation is used to identify and select the most relevant training samples from the candidate pool for adaptation. In particular, the fingerprints of the test points are predicted (starting only from spectrum and chemical formula as input) and K-means clustering is performed on them. The selection is performed using one point per cluster at a time, depending on the cosine similarity of the fingerprints logits. The selected samples are then used for gradient updates. This process is repeated until all the clusters have been used to select a new training batch.

## 2 METHODS: TEST-TIME TUNING

Test-time tuning is an adaptation strategy (Sun et al., 2020; Hübotter et al., 2025) that refines a model during prediction by leveraging unlabeled test points (Gammerman et al., 1998; Farahani et al., 2021). Unlike standard training, the algorithm does not rely solely on knowledge acquired during pre-training. Instead, it uses the unlabeled test points to select the most relevant samples from a large labeled dataset — typically the training set, here referred to as the candidate pool. Only these selected samples are then used to update the model. Since they contain the most relevant information for the considered test point, the model's ability to predict the unknown target is improved.

Our goal is to recover the molecular structure of an unknown compound from its MS/MS spectrum, where the SMILES string is the unknown target label. A natural selection strategy would be to retrieve the candidates with most similar spectra using nearest-neighbor search. However, MS/MS spectra from the same molecule can vary significantly depending on factors such as collision energy, instrument type, and adduct, making spectral similarity an unreliable proxy for structure elucidation. Instead, our selection relies on approximate molecular structure, rather than on spectral information. For each spectrum in both the candidate pool and the test set, the model predicts a 128-bit Morgan fingerprint. Candidate points whose fingerprints have the highest cosine similarity to a given test point are then retrieved via nearest-neighbor search.

In practice, experiments are typically performed in batches, so the full test set is assumed to be available upfront. To handle large test sets efficiently, the predicted test fingerprints are clustered via K-means, and one representative per cluster is processed at a time, as shown in Figure 2. For each representative, a fixed-size batch of training samples is retrieved from the candidate pool and used for a one-step gradient update before moving to the next cluster. By iterating across all clusters, the model gradually adapts to the specific characteristics of the target data, after which evaluation is performed on the full test set. Dataset-specific batch sizes and number of clusters were selected via ablation studies (Figure 9 and Table 9). See Sections B.8 and B.9 for more details.

## 3 RESULTS

### 3.1 CONSISTENT IMPROVEMENTS OVER *de-facto* STATE-OF-THE-ART METHODS

We benchmark our approach comparing test-time tuning (TTT), standard fine-tuning (FT) and pre-training (PT) performances on the NPLIB1 (Dührkop et al., 2021) and MassSpecGym (Bushuiev et al., 2024) datasets across multiple evaluation metrics (see Table 1). The proposed framework delivers consistent improvements in Top–$k$ accuracy compared to existing models, highlighting its ability to generalize across diverse datasets and domain conditions. In particular, state-of-the-art performances are achieved on both datasets, with a Top–1 accuracy of 12.88% on NPLIB1 and 3.16% on MassSpecGym, on which all the baselines presented in (Bushuiev et al., 2024) only obtained 0%. For completeness, we make a distinction between the original NPLIB1 dataset proposed by (Dührkop et al., 2021), which we refer to as NPLIB1-Full, and a smaller version, here named NPLIB1-DiffMS, obtained by following the preprocessing procedure proposed in Bohde et al. (2025) (see Section A for further details). Our framework also surpasses the de-facto state-of-the-art DiffMS, with a relative gain of 67% in Top–1 accuracy on NPLIB1-DiffMS and 27% on MassSpecGym-DiffMS, with Top–1 accuracy of 13.95% and 2.93% respectively.
Beyond accuracy, the Tanimoto similarity and Maximum Common Edge Subgraph (MCES) distance in Table 1 also demonstrate that our approach consistently generates candidates that are chemically closer to the ground truth, providing richer structural insights and supporting more informed decision-making during the elucidation process than the existing counterparts. An in-depth analysis of the predictions is presented in Section 3.7.

In many analytical and predictive workflows, newly acquired data, such as newly measured MS/MS spectra, are measured without substantial prior knowledge of the underlying chemical distribution. Consequently, it is inherently unclear whether a given test instance lies within the distribution represented in the model's training data or departs substantially from it. This uncertainty motivates the development of methods capable of performing reliably under both conditions, when test data are in-distribution (i.e., sufficiently similar to training examples) and when they are out-of-distribution. In Sections 3.2 and 3.3 we demonstrate that the proposed test-time tuning framework is robust across both regimes, while comparing with conventional fine-tuning.

| MODEL | Top–1 | | | Top–10 | | |
|---|---|---|---|---|---|---|
| | ACCURACY ↑ | MCES ↓ | TANIMOTO ↑ | ACCURACY ↑ | MCES ↓ | TANIMOTO ↑ |
| NPLIB1-FULL | | | | | | |
| This work (TTT) | 12.21% | 6.65 | 0.59 | 28.80% | 4.67 | 0.74 |
| This work (FT) | 12.42% | 6.40 | 0.61 | 30.83% | **4.42** | **0.76** |
| **This work (TTT-FT)** | **12.88%** | **6.28** | **0.62** | **31.32%** | 4.42 | 0.76 |
| NPLIB1-DIFFMS | | | | | | |
| Spec2Mol (Litsa et al., 2023)‡ | 0.00% | 27.82 | 0.12 | 0.00% | 23.13 | 0.16 |
| MADGEN (Wang et al., 2025)† | 1.0% | 70.45 | - | 1.0% | 45.64 | - |
| MIST + Neuraldecipher (Goldman et al., 2024b; Le et al., 2020)‡ | 2.32% | 12.11 | 0.35 | 6.11% | 9.91 | 0.43 |
| MIST + MSNovelist (Goldman et al., 2024b; Stravs et al., 2022)‡ | 5.40% | 14.52 | 0.34 | 11.04% | 10.23 | 0.44 |
| DiffMS (Bohde et al., 2025)‡ | 8.34% | 11.95 | 0.35 | 15.44% | 9.23 | 0.47 |
| This work (FT) | 13.20% | 5.77 | **0.64** | 31.14% | 3.93 | **0.77** |
| **This work (TTT)** | **13.95%** | **6.07** | 0.63 | **31.38%** | **3.96** | 0.77 |
| MASSSPECGYM | | | | | | |
| SMILES Transformer (Sennrich et al., 2016; Weininger, 1988)* | 0.00% | 79.39 | 0.03 | 0.00% | 52.13 | 0.10 |
| SELFIES Transformer (Krenn et al., 2020)* | 0.00% | 38.88 | 0.08 | 0.00% | 26.87 | 0.13 |
| Random Generation (Bushuiev et al., 2024)* | 0.00% | 21.11 | 0.08 | 0.00% | 18.26 | 0.11 |
| **This work (TTT)** | **3.16%** | **11.77** | **0.46** | **6.07%** | **9.65** | **0.54** |
| MASSSPECGYM-DIFFMS | | | | | | |
| MIST + MSNovelist (Goldman et al., 2024b; Stravs et al., 2022)‡ | 0.00% | 45.55 | 0.06 | 0.00% | 30.13 | 0.15 |
| Spec2Mol (Litsa et al., 2023)‡ | 0.00% | 37.76 | 0.12 | 0.00% | 29.40 | 0.16 |
| MIST + Neuraldecipher (Goldman et al., 2024b; Le et al., 2020)‡ | 0.00% | 33.19 | 0.14 | 0.00% | 31.89 | 0.16 |
| MADGEN (Wang et al., 2025)† | 0.8% | 74.19 | - | 1.6% | 53.50 | - |
| DiffMS (Bohde et al., 2025)‡ | 2.30% | 18.45 | 0.28 | 4.25% | 14.73 | 0.39 |
| **This work (TTT)** | **2.93%** | **11.81** | **0.46** | **5.51%** | **9.75** | **0.53** |

Table 1: De novo structural elucidation performance on NPLIB1 (Dührkop et al., 2021) and MassSpecGym (Bushuiev et al., 2024) datasets, and DiffMS versions of the same (limited to compounds containing only carbon, oxygen, nitrogen, hydrogen, chlorine, fluorine, sulfur or phosphorus atoms and spectra obtained with $H^+$ adduct). The best performing model for each metric is highlighted in bold.

* Baselines for de novo molecule generation challenge on MassSpecGym (Bushuiev et al., 2024).
† Result taken from the respective indicated paper.
‡ Results of baseline approaches implemented within DiffMS, taken from (Bohde et al., 2025). We assume all the models were evaluated on NPLIB1-DiffMS version of the dataset.

## 3.2 IN-DISTRIBUTION REGIME: TEST-TIME TUNING ACHIEVES FINE-TUNING PERFORMANCE

We analyze here the case where the target spectra are generated by molecules that closely resemble those included in the training set. In this in-distribution setting, fine-tuning on the experimental training set is highly effective for achieving optimal performance on the test set, since all the available data can be considered informative for the model to learn characteristics relevant to the test set. Consequently, standard fine-tuning serves as an upper bound for any improvements achievable through test-time tuning, as shown in Figure 3a (top-right), where fine-tuning (blue) is represented as the asymptotic limit to test-time tuning (orange).

A case study for this scenario is NPLIB1, where the test set is a hold-out from the same dataset, and the train set contains molecules similar to the ones in the test set. This is proven by Figure 3a (bottom), where the distribution of the Tanimoto similarity of test molecules to the train molecules is skewed towards one, and the distribution of the number of heavy atoms in train, test and validation split overlap consistently. We show in Table 2 that the fine-tuned model on NPLIB1-Full reaches Top–1 accuracy of 12.42%, while the test-time tuned model reaches comparable performance at 12.21% Top–1 accuracy. Interestingly, when test-time tuning is applied, the choice of starting from the pre-trained model or the fine-tuned one does not severely affect performance. By looking at the results in Table 2, a relative gain of only 5% is observed when starting from the fine-tuned model instead of from the pre-trained one, rendering prior fine-tuning negligible.

The upper bound imposed by standard fine-tuning can be meaningfully surpassed only by leveraging additional data, expanding the candidate pool with more experimental spectra, as shown in green Figure 3a (top). In the case of NPLIB1-Full, the result is shown in the last (underlined) row of the top section of Table 2, reaching a Top–1 accuracy of 17.28%. Refer to Section 3.6 for a more detailed description and analysis.

## 3.3 OUT-OF-DISTRIBUTION REGIME: TEST-TIME TUNING ENABLES DOMAIN ADAPTATION

In situations where the acquired spectra originate from molecular structures that are poorly represented or entirely diverse from those in the training data, the problem shifts to an out-of-distribution regime. In out-of-distribution cases, the heterogeneity of the spectra used for training and across the

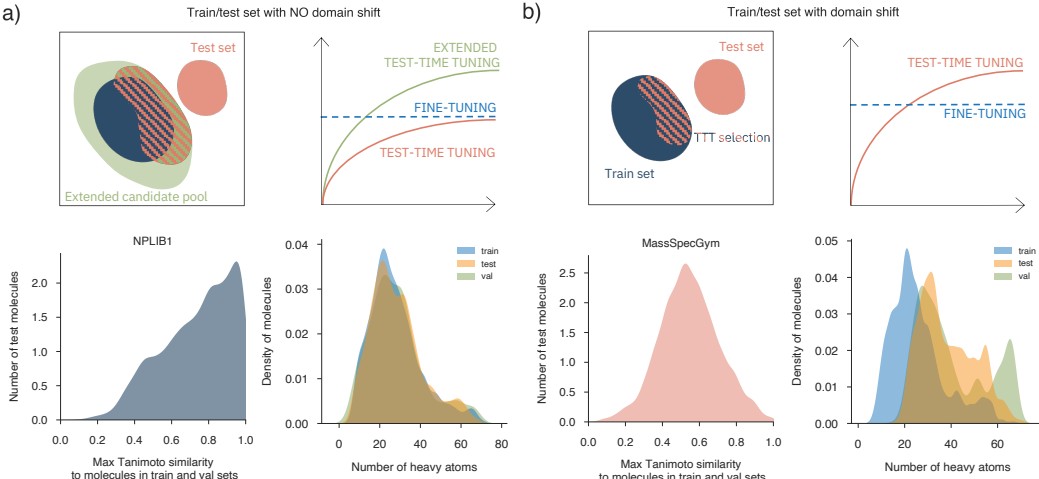

Figure 3: Overview of tuning strategies under varying domain conditions, accompanied by representative datasets. (a) No domain shift: when the training and test sets share the same distribution, both fine-tuning and test-time tuning yield comparable performance. In such cases, fine-tuning typically represents the upper bound of achievable performance. However, performance can be further enhanced through extended test-time tuning, which leverages additional data (green) to expand the candidate pool. The NPLIB1 dataset exemplifies this scenario. The second row illustrates the distribution of test molecules based on their maximum Tanimoto similarity to molecules in the training and test sets (left), as well as the distribution of molecules across the training, test, and validation sets based on their number of heavy atoms (non-hydrogen). The analysis is performed on NPLIB1-Full. (b) Domain shift: in scenarios where the training and test sets differ significantly, fine-tuning may lead to degraded performance. In contrast, test-time tuning dynamically selects relevant samples, thereby improving generalization to the target distribution. The MassSpecGym benchmark serves as an example of this condition, with corresponding distributions shown in the second row.

observed molecules is so marked that some training points are only weakly informative for the target spectra being analyzed at test time. Consequently, fine-tuning distracts the model during adaptation and leads to catastrophic forgetting of previously learned representations, thus degrading performance. Our results show that test-time tuning is a robust alternative approach in these scenarios.

In MassSpecGym, train/test splits are constructed using the MCES distance, ensuring that no similar molecules are shared between train and test set. As Figure 3b (bottom) illustrates, the train and test sets follow different distributions, making domain adaptation strategies essential to bridge the gap between source and target domains. Fine-tuning on the entire MassSpecGym training set results in a performance drop compared to the pre-trained model, with Top–1 accuracy decreasing from 1.89% to 1.13% (c.f. Table 2). This indicates that using all available training data does not enhance learning; instead, it causes the model to forget knowledge acquired during pre-training. In contrast, applying test-time tuning yields a relative improvement of 67%, reaching a Top–1 accuracy of 3.16%, which is a notable improvement if compared to the baselines proposed in (Bushuiev et al., 2024). These results suggest that the training set does contain informative data points, but their effective use requires selective and adaptive strategies rather than broad fine-tuning. This highlights the importance of methods that can dynamically adapt to the target distribution without sacrificing previously learned knowledge.

## 3.4 SIMULATION-BASED PRE-TRAINING SUBSTANTIALLY ENHANCES MODEL PERFORMANCE

Results demonstrate that our model can effectively learn from simulated spectroscopic data from Alberts et al. (2024b), leading to marked improvements over training on experimental data alone. This highlights the value of simulation-based pre-training as a scalable solution to the limited availability of high-quality MS/MS experimental spectra, despite the observable differences between the two signals (see Figure 5).

| | Top–1 Accuracy ↑ | Top–5 Accuracy ↑ | Top–10 Accuracy ↑ | Valid SMILES ↑ |
|---|---|---|---|---|
| NPLIB1-FULL | | | | |
| Fine-tuning from scratch | 0.62% | 1.98% | 2.09% | 11.86% |
| Zero-shot PT model | 3.84% | 7.86% | 9.17% | 74.32% |
| Fine-tuning PT model | 12.42% | 25.91% | 30.83% | 88.01% |
| Test-time tuning (from PT model) | 12.21% | 24.68% | 28.80% | 87.19% |
| **Test-time tuning from FT model** | **12.88%** | **26.38%** | **31.32%** | **89.12%** |
| Extended Test-time tuning (from PT model) | 17.28% | 27.72% | 31.74% | 84.68% |
| MASSSPECGYM | | | | |
| Fine-tuning from scratch | 0.00% | 0.01% | 0.02% | 14.44% |
| Zero-shot PT model | 1.89% | 3.79% | 4.39% | 62.72% |
| Fine-tuning (from PT model) | 1.13% | 2.46% | 2.93% | **67.80%** |
| Test-time tuning from FT model | 1.26% | 2.60% | 3.22% | 67.76% |
| **Test-time tuning (from PT model)** | **3.16%** | **5.39%** | **6.07%** | 64.31% |

Table 2: Performances of the fine-tuned and test-time tuned models on the experimental datasets NPLIB1-Full and MassSpecGym. To highlight the impact of simulated data, the performances of the model trained from scratch on the experimental datasets are shown, as well as the zero-shot evaluation of the pre-trained model. Underlined the results of test-time tuning strategy on NPLIB1, when the candidate pool is extended with additional experimental data (from MassSpecGym).

Pre-training on simulated spectra leads to marked improvements in zero-shot performance, reaching 3.84% Top–1 accuracy on NPLIB1-Full against 0.62%, as shown by the first and second rows of Table 2. Performance on MassSpecGym also improved, rising from 0% to 1.89%. Although modest, the gain is noteworthy given the performances of almost all the current models being around 0% (Bushuiev et al., 2024), highlighting the difficulty of this benchmark. In both cases, the fraction of valid SMILES among the Top–10 predictions by the model trained without simulations is very low, indicating that the experimental data alone from both datasets are insufficient for learning chemically consistent representations from scratch. In contrast, adding simulated spectra in the pre-training stage provides a clear and substantial boost, addressing the limitations identified above.

## 3.5 GENERALIZATION TO COMPLETELY UNSEEN COMPOUNDS

We assess the generalization capabilities of the proposed framework, namely its ability to elucidate the structure of compounds never encountered during training in any form. We evaluate the model after removing from the simulated pre-training data all spectra whose SMILES appear in the experimental test sets (955 and 640 simulated spectra corresponding to molecules in MassSpecGym and NPLIB1 test set respectively). This analysis directly probes the model's ability to elucidate structures of genuinely novel molecules, with no structural overlap between the pre-training data and the evaluation set. It is important to note that in this setting, the model has never been exposed, neither through simulated nor experimental spectra, to any molecule present in the test set. This represents the most challenging out-of-distribution condition: the model must predict the structure of entirely unknown compounds whose spectra and SMILES have never been observed in any form during training. A model that merely memorizes pre-training data rather than learning generalizable structure-spectrum relationships would achieve 0% accuracy in this setting. Results in Table 3 show that test-time tuning retains consistent performance even under these strict conditions, demonstrating the framework's potential to elucidate the structure of truly novel compounds and confirming that the reported gains reflect genuine generalization rather than memorization of pre-training data.

| GENERALIZATION TO COMPLETELY UNSEEN COMPOUNDS | | | | |
|---|---|---|---|---|
| | Top–1 Accuracy ↑ | Top–5 Accuracy ↑ | Top–10 Accuracy ↑ | Valid SMILES ↑ |
| NPLIB1-FULL | | | | |
| Fine-tuning PT model | 10.46% | **23.98%** | **28.46%** | **92.45%** |
| Test-time tuning (from FT model) | **10.54%** | 22.90% | 26.82% | 83.17% |
| MASSSPECGYM | | | | |
| Fine-tuning (from PT model) | 0.73% | 1.86% | 2.36% | **67.41%** |
| **Test-time tuning (from PT model)** | **1.28%** | **2.92%** | **3.46%** | 63.26% |

Table 3: Performances of fine-tuned and test-time tuned models performing the best on NPLIB1-Full and MassSpecGym, starting from the pre-trained model on the dataset of simulated data after removing the spectra obtained from SMILES appearing in the experimental test sets.

### 3.6 Leveraging additional experimental spectra improves test-time adaptation on NPLIB1

We examine the effect of incorporating additional experimental spectra into the candidate pool of the test-time tuning of NPLIB1. We expand the candidate pool of NPLIB1-Full by combining its original training set with MassSpecGym. This extension leads to a substantial improvement in Top–1 accuracy, going from 12.21% when using only the NPLIB1 training set, to 17.28% after adding the extra spectra, equal to 41% relative gain (cf. Table 2). This demonstrates that the method is able to select training points containing relevant structural information for the test set, even from a large candidate pool of hundreds of thousand spectra.

### 3.7 Improved molecular similarity and structural accuracy of predictions

Even when the model fails to predict the correct molecular structure, its predictions still provide relevant guidance toward structurally relevant candidates, which can aid chemists. The Tanimoto similarity and MCES distance can be seen as proxies for how two molecules are closely related to each other, and hence how informative a predicted structure may be to a chemist when narrowing down the search space for the true structure. The higher the similarity, the closer the predicted structure is to the actual target, and, inversely, the smaller the distance, the easier it should be to converge to the true structure.

For all the datasets in Table 1, the Tanimoto similarity is higher for the predictions of our framework than those of other models. The average Tanimoto similarities of the Top–1 predicted molecules (excluding the invalid SMILES) are 0.62 on NPLIB1-Full and 0.46 on MassSpecGym, hence meaningful matches on average. Similarly, the MCES distance is lower for our approach than the baselines, with MCES reaching 6.28 on NPLIB1-Full and 11.77 on MassSpecGym. Results are similar on NPLIB1-DiffMS and MassSpecGym-DiffMS, for which both metrics improve even further.

Taking inspiration from the analysis done in Bohde et al. (2025), we show Table 4, where we classified the molecules depending on their Tanimoto similarity with the respective target. More precisely, two classes are introduced: a meaningful match is defined if Tanimoto similarity $\geq 0.4$, while a close match in case Tanimoto similarity $\geq 0.675$. While meaningful matches indicate general structural correctness, close matches reflect near-identical chemical similarity, which is more challenging to achieve. The proposed approach outperforms existing methods in both classes, achieving nearly 350% increase in Top–1 meaningful matches (56.22%) compared to DiffMS (12.41%). These results underscore the model's strength in generating highly accurate structures, not just broadly correct ones.

To conclude, Figure 4 presents an example of predicted molecules for a given target. Although the model does not identify the exact molecule on its first attempt, it succeeds on the second. Interestingly, the first three predictions are stereoisomers with a Tanimoto similarity of 1.0, highlighting the model's strong understanding of the target structure.

### 3.8 Excluding stereochemistry aligns evaluation with MS/MS spectral information content

Since fragmentation patterns are generally invariant to stereoisomeric differences, MS/MS spectra do not encode stereochemical information and the model cannot be expected to distinguish stereoisomers from spectral input alone. Consistent with this, Figure 4 already shows that the model's top predictions are often stereoisomers of the correct structure with Tanimoto similarity of 1.0. To reflect this inherent limitation of the data, we additionally evaluate our method by removing stereochemical information from predictions and target structures as a post-processing step. Results are reported in Table 5. Valid SMILES rates are unaffected by stereochemistry removal, confirming that the gains are purely due to stereoisomeric equivalences. The improvement is more pronounced on MassSpecGym going from 3.16% to 4.06% Top–1 accuracy, with a relative gain of $\sim$28%. While for NPLIB1-Full, performances improve from 12.88% to 13.09%, accounting for a smaller relative gain of $\sim 2$%. This result is consistent with the higher stereoisomeric diversity expected in the out-of-distribution MassSpecGym test set.

| MODEL | Top–1 | | Top–10 | | Valid SMILES ↑ |
|---|---|---|---|---|---|
| | Meaningful match ↑ (≥ 0.4) | Close match ↑ (≥ 0.675) | Meaningful match ↑ (≥ 0.4) | Close match ↑ (≥ 0.675) | |
| NPLIB1-FULL | | | | | |
| This work (FT) | 72.01% | 43.14% | **89.00%** | **62.41%** | 88.01% |
| This work (TTT) | 70.12% | 38.71% | 88.34% | 58.94% | 87.19% |
| **This work (TTT-FT)** | **72.51%** | **43.81%** | 88.62% | 62.38% | **89.11%** |
| NPLIB1-DIFFMS | | | | | |
| Spec2Mol (Litsa et al., 2023)‡ | 0.00% | 0.00% | 0.00% | 0.00% | 66.5% |
| MIST + Neuraldecipher (Goldman et al., 2024b; Le et al., 2020)‡ | 29.30% | 7.33% | 41.39% | 12.82% | 91.11% |
| MIST + MSNovelist (Goldman et al., 2024b; Stravs et al., 2022)‡ | 32.90 % | 11.78% | 44.79% | 19.02% | 98.60% |
| DIFFMS (Bohde et al., 2025)‡ | 27.40 % | 12.83% | 46.45% | 22.04 % | **100.0%** |
| This work (FT) | **75.00%** | **48.13%** | 90.11% | 65.37% | 90.24% |
| **This work (TTT)** | 72.60% | 43.99% | **91.50%** | **65.45%** | 89.28% |
| MASSSPECGYM | | | | | |
| **This work (TTT)** | **55.53%** | **9.99%** | **76.07%** | **17.03%** | **64.31%** |
| MASSSPECGYM-DIFFMS | | | | | |
| Spec2Mol (Litsa et al., 2023)‡ | 0.0% | 0.0% | 0.0% | 0.0% | 68.5% |
| MIST + Neuraldecipher (Goldman et al., 2024b; Le et al., 2020)‡ | 0.29% | 0.01% | 0.39% | 0.09% | 81.78% |
| MIST + MSNovelist (Goldman et al., 2024b; Stravs et al., 2022)‡ | 0.66% | 0.00% | 1.92% | 0.00% | 98.58% |
| DIFFMS (Bohde et al., 2025)‡ | 12.41% | 3.78% | 32.47% | 6.73% | **100.0%** |
| **This work (TTT)** | **56.22%** | **9.46%** | **76.17%** | **16.93%** | 64.11% |

Table 4: Additional evaluation of the similarity of predicted molecules to the target molecule. Different classes are defined depending on the Tanimoto similarity, as reported at the top of the table and explained in Section 3.7. Percentage of valid SMILES on the first 10 predicted molecules for our models, and over all the predictions for the others (see Table 4 in Appendix in Bohde et al. (2025)). Highlighted in bold the name of the model holding the best Top–1 accuracy on the left, while the best score in each class on the right.

‡ Results of baseline approaches implemented within DiffMS, taken from Bohde et al. (2025). We assume all the models were evaluated on the same NPLIB1 filtered version that we named NPLIB1-DiffMS.

Figure 4: Top–10 predictions for a molecule in MassSpecGym test set. Respective Tanimoto similarity and MCES distance from the target structure are provided below every prediction. The model generates three stereoisomers among the first predictions, indicating structural awareness, but fails to identify the correct SMILES at the first attempt. The correct structure appears as the second candidate (highlighted in green), which positively contributes to Top–5 and Top–10 accuracy. This result motivated the analysis in Section 3.8.

# 4  DISCUSSION

We demonstrate that transformer-based language models, combined with test-time tuning, offer a promising direction for advancing de novo molecular structure elucidation from MS/MS spectra. By eliminating intermediate steps such as fragment annotation, our approach achieves true end-to-end generation of SMILES strings from spectra and chemical formulae. This design not only simplifies the pipeline but also improves the understanding of chemical structures, as evidenced by the high structural similarity of predicted candidates even when exact matches are not obtained.

Test-time tuning proves robust across different distributional regimes. In domains with minimal distribution shift, such as NPLIB1, it achieves comparable results to classical fine-tuning, confirming that performance is not sacrificed even in easier settings. In contrast, in highly heterogeneous settings such as MassSpecGym, test-time tuning becomes essential to recover performance otherwise lost with naive broad adaptation. This flexibility highlights the robustness of our approach and its

|  | Top–1 Accuracy ↑ | Top–5 Accuracy ↑ | Top–10 Accuracy ↑ | Valid SMILES ↑ |
|---|---|---|---|---|
| NPLIB1-FULL | | | | |
| With stereochemistry (TTT-FT) | 12.88% | 26.38% | 31.32% | 89.12% |
| **Without stereochemistry (TTT-FT)** | **13.09%** | **26.53%** | **31.61%** | 89.12% |
| MASSSPECGYM | | | | |
| With stereochemistry (TTT) | 3.16% | 5.39% | 6.07% | 64.31% |
| **Without stereochemistry (TTT)** | **4.06%** | **6.16%** | **6.89%** | 64.31% |

Table 5: Performances of the best performing models on NPLIB1-Full and MassSpecGym with and without stereochemical information removed as a post-processing step. Results with stereochemistry are taken from Table 2.

ability to adapt to diverse data conditions without sacrificing previously learned knowledge.
Moreover, the inclusion of more experimental spectra can provide richer structural information, potentially enabling the model to learn more effectively and significantly enhance performance, particularly in scenarios where the original training set is small, as in NPLIB1 case. This suggests that curating and integrating additional experimental data should be a priority for improving accuracy. Our experiments also indicate that having access to a larger pool of experimental data, increases the likelihood of selecting informative samples during adaptation, which in turn improves generalization and chemical plausibility of predictions.

The impact of simulated data is particularly noteworthy. Pre-training on large-scale simulations provides the model with a richer understanding of fragmentation patterns and structural relationships. It also reduces the limitations imposed by scarce experimental data and sets the stage for more effective fine-tuning and adaptation. The analysis on generalization to completely unseen compounds further supports this: removing simulated spectra whose SMILES overlap with the experimental test sets doesn't result in zero accuracy, meaning that the model learns meaningful spectrum-molecule representations that can be transferred to genuinely unseen compounds, rather than exploiting structural familiarity as a shortcut. Even when imperfect, simulated data are a valuable and scalable resource that should be exploited as broadly as possible to complement scarce experimental measurements. Finally, even when the predicted SMILES does not perfectly match the ground truth, the generated candidates remain chemically meaningful. High Tanimoto similarity and low MCES distances indicate that these predictions provide valuable structural hints, significantly narrowing the search space for human experts. This property transforms the model from a mere predictor into a practical assistant for structure elucidation. Notably, since MS/MS spectra do not encode stereochemical information, evaluating predictions without stereochemistry yields an improvement in performances, further underlining the chemical relevance of the generated candidates.

While the proposed framework demonstrates strong performance across datasets and distributional regimes, it also presents a few practical limitations. The primary challenge concerns scalability: as the candidate pool grows, identifying the most informative spectra for test-time tuning becomes increasingly demanding. Approximate nearest-neighbor or vector-search methods could help mitigating this. See Appendix E for runtime details. More fundamentally, the method requires at least some spectra obtained from structurally relevant compounds in the candidate pool. Performance degrades when informative neighbors are absent or when spectra are noisy. Test-time tuning is highly effective when relevant and reliable information exists, but its benefits diminish when the available data contain no actionable cues. Finally, results are reported for single runs; assessing variance across seeds remains an avenue for future work.

Nevertheless, an important advantage of test-time tuning lies in its efficiency: unlike conventional fine-tuning, it requires substantially fewer data points for adaptation. When the selection of informative spectra can be performed rapidly, this property translates into faster overall adaptation without sacrificing predictive performance, making it particularly appealing in scenarios where computational resources or time are limited.

As the first application of test-time tuning to structure elucidation from spectroscopic data to the best of our knowledge, the results highlight the potential of adaptive language models to transform MS/MS-based structure elucidation workflows. By leveraging test-time tuning and simulated data, these models offer a scalable and flexible solution for navigating chemical diversity, paving the way for more accurate and efficient identification of unknown compounds in metabolomics, natural product discovery, and beyond.

REPRODUCIBILITY STATEMENT

All the data used in this study are public. Links for the download and pre-processing procedure are provided in Appendix A. The code used in this study is open-source (MIT license) and can be found as a GitHub repository at `https://github.com/rxn4chemistry/MultimodalAnalytical/tree/ttt-msms`.

ACKNOWLEDGEMENTS

This publication was created as part of NCCR Catalysis (grant number 225147), a National Centre of Competence in Research funded by the Swiss National Science Foundation.

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

## A   Data

We use three different datasets of MS/MS spectra generated via positive electrospray technique, relying on $H^+$ and $Na^+$ adducts. Simulated MS/MS spectra are obtained from (Alberts et al., 2024b), which sums up to a total of 3 971 930 simulations combining CFM-ID 4.0 (Wang et al., 2021) with collision energy equal to $10 \ eV$, $20 \ eV$ and $40 \ eV$, ICEBERG (Goldman et al., 2024a), and SCARF (Goldman et al., 2023a). An example of the spectra obtained with these techniques is shown in Figure 5. As for the experimental spectra, we used NPLIB1, which is derived from GNPS (Wang et al., 2016) and was first introduced in (Dührkop et al., 2021), and MassSpecGym (Bushuiev et al., 2024), which already provides a fixed train, validation and test split to benchmark against. These datasets contain respectively 19 687 and 231 104 spectra.

However, to allow a fair comparison with results from other methods, we also evaluated different versions of the same datasets. More specifically, the pre-processing applied in Bohde et al. (2025) is performed, consisting in an adduct-based and element-based filtering operation. Only spectra containing $H^+$ adducts, and molecules containing only carbon, oxygen, nitrogen, hydrogen, phosphorus, sulfur, chlorine and fluorine are retained. We call these versions of the datasets MassSpecGym-DiffMS, with 227 341 spectra, and NPLIB1-DiffMS with 7 947 spectra. More details, also about reproducibility, can be found for each dataset in the paragraphs below.

When looking at the spectra obtained for a specific molecule, the simulated ones significantly differ from the ones obtained through experiments, as can be seen in Figure 5, which underscores the challenge of bridging the gap between simulated and experimental spectra during model training and adaptation.

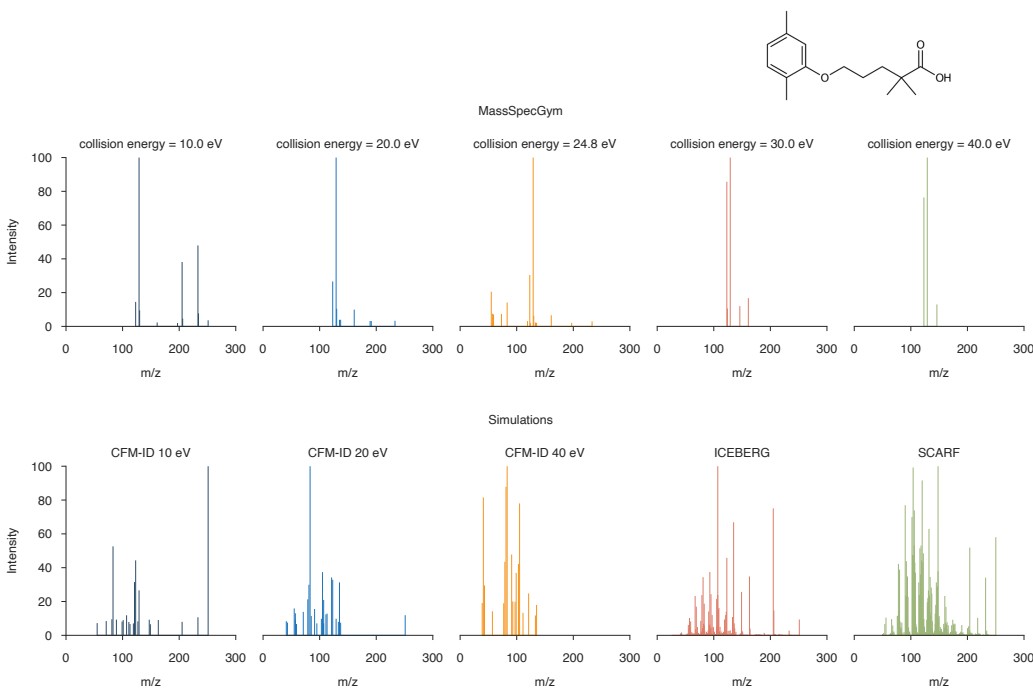

Figure 5: Example of available spectra for one specific molecule in the datasets used in the present work. In this case MassSpecGym (Bushuiev et al., 2024) contains 5 different spectra obtained using different values for the collision energy. In the last row are presented the spectra obtained with the 5 simulation techniques used in (Alberts et al., 2024b) and mentioned above. As it is possible to see they all severely differ from each other.

**Simulated dataset** The pre-training of the model was performed using the dataset of simulated MS/MS spectra from Alberts et al. (2024b). It contains 794 386 entries, each describing a SMILES-MS/MS spectra tuple. Every entry indeed contains 5 different spectra obtained using different sim-

ulation modalities, namely CFM-ID 4.0 (Wang et al., 2021) with collision energy equal to $10\ eV$, $20\ eV$ and $40\ eV$, ICEBERG (Goldman et al., 2024a) and SCARF (Goldman et al., 2023a). As shown in Figure 5. In total, the dataset contains 3 971 930 spectra. The count of unique SMILES is 789 328. The spectra have been normalized to have maximum intensity equal to 100, afterwards, all the peaks with intensity smaller than 1 have been removed, since considered noise. After this pre-processing step every spectrum contains at most 31 peaks for the three CFM-ID modalities, 300 for the SCARF modality, while 595 for ICEBERG.

**MassSpecGym**   To the best of our knowledge, it is the most challenging dataset for MS/MS data, obtained from Bushuiev et al. (2024), it contains 231 104 spectra for 31 602 unique compounds (after canonicalization). Different instruments were used for the collection of these spectra, such as varying collision energy and $H^+$ and $Na^+$ adducts. Train, validation and test splits are given, since they were created using a threshold of 10 on the MCES distance between the molecules in train, validation and test set, which is what makes the dataset so challenging. In particular, the test set contains 17 556 spectra. It can be downloaded from `https://huggingface.co/datasets/roman-bushuiev/MassSpecGym/blob/main/data/MassSpecGym.tsv`. To avoid confusion, we refer to this original version of the dataset as just MASSSPECGYM.

In fact, another version of the dataset is introduced in Bohde et al. (2025), where a filtering on the elements contained in the molecules is performed. More precisely, only compounds with carbon, oxygen, nitrogen, hydrogen, phosphorus, sulfur, chlorine and fluorine elements were kept, for a total of 30 640 unique compounds, 227 341 spectra, of which only 17 082 in the test set (instead of the initial 17 556 spectra). See Table 6 for more details. We refer to this filtered version of MassSpecGym as MASSSPECGYM-DIFFMS.

**NPLIB1**   The second experimental dataset we used is NPLIB1, which was derived from GNPS library (Wang et al., 2016) and MassBank (Horai et al., 2010), and firstly introduced in Dührkop et al. (2021), where it was used for the SVM training (and not to be confused with the structures dataset instead used for CANOPUS training[1]). It can be downloaded from `https://bio.informatik.uni-jena.de/wp-content/uploads/2020/08/svm_training_data.zip`, while a full description of the data they used is available at `https://bio.informatik.uni-jena.de/data/`. The folder contains 10 709 spectra files (see 'compound_ids.tsv' file in the downloaded folder) for a total of 19 687 spectra, since multiple spectra with different collision energies were measured for several compounds, and 8553 unique compounds. The spectra were obtained using both $H^+$ and $Na^+$ adducts. More details can be found in Table 6. We refer to this dataset as NPLIB1-FULL.

This dataset has been used in later works, such as Goldman et al. (2023b; 2024b; 2023a) and Bohde et al. (2025), where it was subject to filtering procedures, generating several different versions of the dataset. First of all, MIST tool (Goldman et al., 2023b), which can be used to annotate tandem mass spectra peaks with chemical structures, was trained on a dataset composed by both commercially and publicly available data. The public dataset is composed by the NPLIB1 spectra obtained by using only $H^+$ adduct. After this adduct-based filtering, only 8030 spectra files remain, which contain a total of 14 532 spectra with different collision energies, corresponding to 7131 unique compounds[2]. The same dataset was then also used in Goldman et al. (2024b). However, all these works mainly addressed spectra predictions tasks instead of structure elucidation, as done in this work, and they processed the dataset by merging all the peaks present in a single spectra file, and obtained under different collision energies, in only one spectra. On the other hand, for our purpose, it is straightforward to separate the peaks obtained at different collision energies in different spectra objects, ending up with multiple spectra for several files. We refer to this dataset as NPLIB1-$H^+$.

In the end, also DiffMS (Bohde et al., 2025) was evaluated on the structure elucidation task for NPLIB1 dataset, however, as for MassSpecGym dataset, they performed an additional filtering

---

[1]For full clarity, a list of spectra files is also available in Supplementary Table 6 of the respective manuscript (Dührkop et al., 2021), which shows 10 710 file names. However, it refers to a slightly different dataset, which was instead used for the training of CANOPUS tool, instead of the SVM presented in the manuscript. Indeed, 4 file names in this list are not present in the SVM training dataset (ids: CCMSLIB00000078787, CCMSLIB00000081065, CCMSLIB00000847829, CCMSLIB00000855420), while 3 different file names were added (ids: CCMSLIB00001058585, CCMSLIB00001058867, CCMSLIB00001059041).

[2]The initial folder containing all the 10 709 spectra files can also be downloaded from `https://zenodo.org/records/8316682`.

on top on the adduct-based one, so that only spectra of molecules containing only carbon, oxygen, nitrogen, hydrogen, phosphorus, sulfur, chlorine and fluorine elements were kept. Resulting with a total of 7947 files, containing 7127 unique compounds. Moreover, they did not merge all the peaks in each file in a unique spectrum as mentioned before for other works, but most likely used only the first spectrum written in every file, ending up with the same number of spectra as the files, equal to 7947, with 6748 in training set, 396 in validation set and 803 in test set. We obtained this dataset by running DiffMS code with the configurations and parameters they provided to reproduce the results. In particular, the train-val-test split used can be found in the file 'DiffMS/data/canopus/splits/canopus_hplus_100_0.tsv'. We additionally evaluate our method on this last version of the NPLIB1 dataset to fairly compare with DiffMS results, and we refer to it as NPLIB1-DIFFMS.

When the splitting is not provided, we divide the data approximately as: 20% test set, 5% validation set and 75% training set, always ensuring different spectra obtained for the same compound falls in the same split (to avoid data leakage).

Notebooks for the generation of all the different versions of the datasets are available in the public code of this work.

We did not discard any spectrum from any dataset, only normalization was applied and noisy peaks removed. More precisely, for every spectrum, the peaks were normalized such that the maximum intensity was equal to 100, consequently, the peaks with intensity smaller than 1 were removed. All the SMILES strings were canonicalized and Morgan fingerprint with size 128 calculated using rdkit (Landrum & Community).

| | Simulations | NPLIB1-Full | NPLIB1-$H^+$ | NPLIB1-DiffMS | MassSpecGym | MassSpecGym-DiffMS |
|---|---|---|---|---|---|---|
| # files | - | 10709 | 8030 | 7947 | - | - |
| # spectra | 3971930 | 19687 | 14532 | 7947 | 231104 | 227341 |
| # unique SMILES | 789328 | 8553 | 7131 | 7053 | 31602 | 30640 |
| Split provided* | ✗ | ✗ | ✗ | ✓ | ✓ | ✓ |
| # train set | 2978947 | 14897 | 10927 | 6748 | 194119 | 191216 |
| # val set | 198597 | 908 | 622 | 396 | 19429 | 19043 |
| # test set | 794386 | 3882 | 2983 | 803 | 17556 | 17082 |
| # unique SMILES in test set | - | 1711 | 1427 | 701 | 3170 | 3076 |
| Adducts | $H^+$ | $H^+$, $Na^+$ | $H^+$ | $H^+$ | $H^+$, $Na^+$ | $H^+$ |
| C,H,O,P,N,S,Cl,F | ✓ | ✓ | ✓ | ✓ | ✓ | ✓ |
| B,Br,I | ✓ | ✓ | ✓ | ✗ | ✓ | ✗ |
| Si | ✓ | ✓ | ✓ | ✗ | ✓ | ✗ |
| Se | ✗ | ✓ | ✓ | ✗ | ✓ | ✗ |
| As | ✗ | ✗ | ✗ | ✗ | ✓ | ✗ |
| Sn,Al | ✗ | ✗ | ✗ | ✗ | ✗ | ✗ |

Table 6: Details about the content of the different datasets used. In particular, the number of spectra files used, the effective total number of spectra (by dividing peaks obtained at different collision energies in different spectra files), number of unique compounds. We also report if the train-val-test split was given, together with the number of spectra in each set. We also checked the elements present in each datasets version.
* When the train-val-test split is not provided, we perform a 75-5-20 split ensuring no spectra for the same compound fall in different sets. If provided, we adopted the given one and simply report the numbers.

# B    METHODS

## B.1    ARCHITECTURE AND MODALITIES

The proposed model follows a sequence-to-sequence encoder-decoder architecture. It takes as input the MS/MS spectrum and the chemical formula of a molecule to predict the corresponding SMILES, as illustrated in Figure 2. We decided to include the chemical formula in the inputs of the model since it is usually known when tandem mass spectroscopy is performed either computationally or during experiments. Its inclusion allows the model to gather more information about the elements present in the target molecule. Every modality, meaning spectrum, chemical formula and SMILES, is treated as text. In particular, the peaks of each spectrum are encoded as a list of tuples of the mass-to-charge ratio and intensity $[mzs, I]$ and converted then to a string.

## B.2 MODEL DETAILS

We train a transformer encoder–decoder with ∼150 million parameters using cross-entropy loss for SMILES generation and an additional binary cross-entropy loss for fingerprint prediction. The model is configured with a hidden dimensionality of 1024 and multi-head attention with 8 heads for both the encoder and decoder. The architecture consists of 6 layers in the encoder and 6 layers in the decoder, each with a feed-forward network dimension of 2048. Normalization strategies include multimodal normalization, final layer normalization, and post-layer normalization, while gated linear units are enabled to improve representational capacity.

## B.3 TRAINING DETAILS

Optimization is performed using the AdamW algorithm (Kingma & Ba, 2017) with $\beta_1$=0.9 and $\beta_2$=0.999, and no weight decay is applied. The exponential learning rate scheduler is used with initial learning rate of $1e^{-4}$ for pre-training and $5e^{-5}$ for fine-tuning and test-time tuning, with a decay factor $\gamma$ of 0.95 for standard pre-training and fine-tuning, and 0.995 for test-time tuning. Batch size was set to 16 for pre-training and to 32 for fine-tuning. Different values depending on the dataset were then used for test-time tuning, see Table 9. Pre-training ran for up to 200 epochs, with early stopping based on validation token accuracy, typically halting around 60 epochs. Fine-tuning followed the same criterion, stopping between 20–30 epochs depending on the dataset. For test-time tuning, the number of epochs is determined by the number of K-means clusters selected.
Teacher forcing technique is implemented for next token prediction (Williams & Zipser, 1989).

## B.4 TOKENIZATION

To convert MS/MS spectra into a format suitable for the transformer model, we tokenize each spectrum as a sequence of peak pairs $[m/z, I]$, see Figure 6. The peaks are then concatenated into a string representation. This tokenized spectrum is combined with the chemical formula of the respective molecule to form the complete input. By treating both the spectrum and formula as text, the model can leverage language modeling techniques for end-to-end SMILES generation.

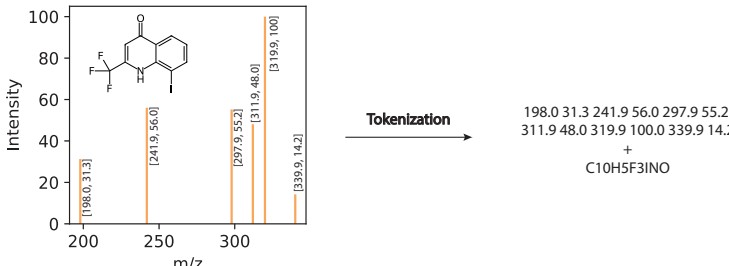

Figure 6: Illustration of the tokenization process for an MS/MS spectrum of the corresponding depicted molecule with formula $C_{10}H_5F_3INO$. Every peak is stored as a tuple of mass-to-charge ratio and intensity, then converted to a simple string and concatenated with the chemical formula.

## B.5 FINGERPRINT ALIGNMENT

An additional component of the model is a multilayer perceptron (MLP) placed on top of the encoder, which predicts molecular fingerprints from the encoder's output embeddings (see Figure 2). To account for this prediction task, a binary cross-entropy loss term is incorporated alongside the standard cross-entropy loss used for the encoder–decoder, as shown in Equation 1. This, often referred to as fingerprint alignment, allows the model to learn representations that better capture chemical information, guiding SMILES predictions toward more plausible structures. Specifically, we use 128-bit Morgan fingerprints, computed using RDKit (Landrum & Community). Stereochemical information is deliberately excluded, as MS/MS fragmentation patterns are generally invariant to stereoisomeric differences, meaning the spectral input itself contains no stereochemical information — an assumption further supported by our results in Section 3.8.

$$\mathcal{L} = \mathcal{L}_{\text{CE, SMILES}} + \lambda\,\mathcal{L}_{\text{BCE,Fingerprints}} \tag{1}$$

## B.6 Formula constrained generation

To further leverage the chemical formula provided as input, we implement formula-constrained generation at prediction time (Alberts et al., 2025c). Since the model generates SMILES strings token by token, we dynamically restrict the set of allowed tokens at each decoding step by removing those that would violate the given chemical formula. This ensures that the generated SMILES is always consistent with the specified formula, thereby increasing the likelihood of producing the correct target structure. Beyond improving validity, this constraint also guides the model toward chemically plausible candidates, enhancing both accuracy and interpretability.

## B.7 Rejection sampling

Alongside formula constraints, we apply rejection sampling to better exploit the model's ability to generate meaningful candidates, while reducing the impact of invalid SMILES predictions. Specifically, the model generates 50 SMILES strings, and we retain the first 10 (or less if there are not enough) that pass chemical syntax and structural validity checks.

## B.8 Test-time tuning

Test-time tuning is a transductive strategy (Gammerman et al., 1998; Farahani et al., 2021; Sun et al., 2020) that updates parameters at inference time using only unlabeled test inputs, contrasting with pre-training/fine-tuning schemes that assume target data are available during training, to better align predictions with the characteristics of the input data. Unlike conventional training, which relies on a fixed dataset, this method exploits information available at inference time to refine the model without requiring full retraining. The process typically involves selecting relevant training points from a large candidate pool, often using a nearest-neighbor strategy (see Figure 2). Test-time tuning is particularly valuable in scenarios involving domain shifts between training and test sets (Hübotter et al., 2025), where labeled data exist only in the source domain but some unlabeled target data can be leveraged during inference —a setting commonly referred to as transductive transfer learning. By dynamically adapting to the test distribution, this strategy improves robustness and predictive accuracy, especially in challenging conditions where distribution shifts would otherwise degrade performance.

## B.9 Selection of relevant data points

In our implementation, we first predict 128-bit Morgan fingerprints from the spectra in test set and candidate pool using an auxiliary MLP head on the transformer encoder. We then perform k-means clustering over the molecular fingerprints of the test set to uniformly sample the test space. Iterating over the clusters, we select one representative test instance per cluster (closest to the centroid) and retrieve a fixed-size batch of candidate training points from the pool based on cosine similarity in fingerprint-logit space. Nearest-neighbor search is used. These training points correspond, if the representation is good enough, to the most relevant data points for the current test instance. Every selected batch is used for one-step gradient updates to the encoder–decoder and fingerprint head, in a sequential way. Embeddings (and fingerprints) are periodically refreshed to reflect updated parameters. Evaluation is performed on the whole test set once every cluster representative has been explored. In Figure 7, we show how, already from the first iterations, the method selects data points from the candidate pool that have fingerprints with cosine similarity closer to the considered test point than the rest of the candidate pool. In addition, the average cosine similarity of the selected points to the test point slightly increases over the iterations.

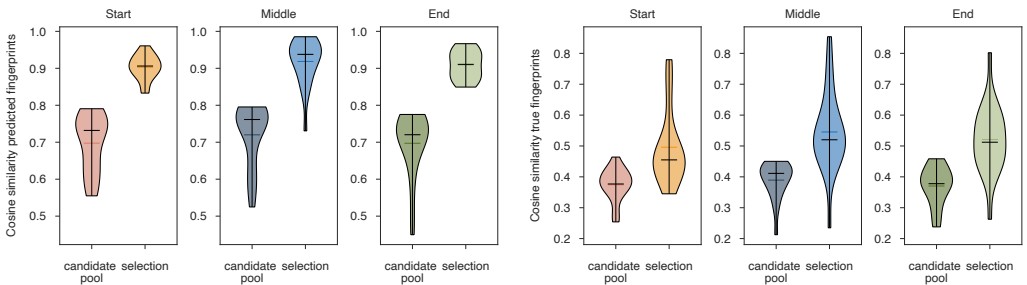

Figure 7: Comparison of the cosine similarity distribution of the predicted (left) and true (right) fingerprints of the candidate pool and selected portion with the considered test point, over different iterations at the beginning, during and at the end of the tuning process.

# C  ADDITIONAL RESULTS

## C.1  ANALYSIS PERFORMANCES BASED ON SPECIFIC SPECTRAL FEATURES

To better understand the factors affecting the performance of the test-time tuned model on MassSpecGym, we analyzed accuracy and structural validity across subsets grouped by key spectral features: adduct type, instrument, parent mass, and collision energy. The results of this analysis are shown in Table 7.

First of all, we divided the dataset depending on the adduct. The majority of spectra (80.12%) were protonated species ($H^+$), for which the model achieved a Top-1 accuracy of 3.92%. In contrast, spectra with sodium adducts ($Na^+$) represented only 19.88% of the data and exhibited much lower accuracy (0.09% Top-1), suggesting that the model struggles with non-protonated species, likely due to their lower representation and more complex fragmentation patterns. The fact that the pre-training simulated data contained only spectra with $H^+$ adduct, definitely contributed as well. Despite this, the model manages to correctly guess a few of them.

Secondly, we analyzed the impact of the instrument used to measure the spectra. The ones acquired on Orbitrap instruments dominated the dataset (83.86%), yielding Top-1 accuracy of 2.86%. QTOF spectra, present with a fraction of 13.81%, showed higher Top-1 accuracy (5.24%), indicating that instrument-specific fragmentation characteristics may influence reconstruction quality. Information about the instrument is missing for a small portion of the data (2.33%). The model performed poorly on these across all metrics.

In Table 7 is also possible to see that performance varied substantially with precursor mass. Compounds in the 400–600 Da range achieved the highest Top-1 accuracy (4.75%), while smaller molecules (mass within 200-400 Da) achieved slightly lower performances (3.32% Top-1), but still higher than the accuracy on the whole dataset (3.16% Top-1). On the other hand, large molecules (m > 600 Da) were almost never reconstructed correctly, with even 0% Top-1, 0.34% valid SMILES for molecules with mass larger than 800 Da. This trend suggests that the model is optimized for mid-range masses, while extreme sizes pose challenges, probably due to increased structural complexity and lower availability. In the end, we also addressed the impact of the collision energy used during the measurement, which shows to strongly impact accuracy. Intermediate energies ($\sim$ 45–105 $eV$) yielded the best results, whereas very low ($<30$ $eV$) or very high ($>105$ $eV$) energies led to lower performance. This indicates that fragmentation richness at moderate energies provides the most informative spectra for structure prediction. The lowest performance is although achieved for the spectra for which we have no information about the collision energy, which constitute a massive part (42.12%) of the dataset, with a Top-1 accuracy of only 1.08%.

Overall, these results highlight that data distribution and fragmentation conditions critically shape model performance, with clear biases toward protonated species, mid-range masses, and spectra acquired under intermediate collision energies.

| | | MASSSPECGYM | | | |
|---|---|---|---|---|---|---|
| | | Portion | Top-1 Accuracy ↑ | Top-5 Accuracy ↑ | Top-10 Accuracy ↑ | Valid SMILES ↑ |
| Adduct | $H^+$ | 80.12% | 3.92% | 6.69% | 7.52% | 72.57% |
| | $Na^+$ | 19.88% | 0.09% | 0.14% | 0.14% | 30.76% |
| Instrument | Orbitrap | 83.86% | 2.86% | 4.9% | 5.52% | 65.06% |
| | QTOF | 13.81% | 5.24% | 9.08% | 10.11% | 63.77% |
| | Unknown | 2.33% | 1.47% | 1.47% | 1.47% | 40.05% |
| Parent mass | $200\,Da < m < 400\,Da$ | 21.96% | 3.32% | 6.2% | 7.42% | 99.74% |
| | $400\,Da < m < 600\,Da$ | 43.06% | 4.75% | 8.29% | 9.18% | 91.08% |
| | $600\,Da < m < 800\,Da$ | 27.83% | 1.39% | 1.66% | 1.72% | 11.32% |
| | $m > 800\,Da$ | 7.15% | 0.0% | 0.0% | 0.0% | 0.34% |
| Collision energy | $E < 15\,eV$ | 2.75% | 2.90% | 5.19% | 5.39% | 68.26% |
| | $15\,eV < E < 30\,eV$ | 17.63% | 3.65% | 6.65% | 7.78% | 88.80% |
| | $30\,eV < E < 45\,eV$ | 11.10% | 5.34% | 9.96% | 10.93% | 84.47% |
| | $45\,eV < E < 60\,eV$ | 6.90% | 6.77% | 11.72% | 13.28% | 84.62% |
| | $60\,eV < E < 75\,eV$ | 14.68% | 4.23% | 7.53% | 8.47% | 90.94% |
| | $75\,eV < E < 90\,eV$ | 1.59% | 7.89% | 15.41% | 16.49% | 81.40% |
| | $90\,eV < E < 105\,eV$ | 1.30% | 6.58% | 8.77% | 10.96% | 82.89% |
| | $E > 105\,eV$ | 1.95% | 4.39% | 4.39% | 4.39% | 65.29% |
| | Unknown | 42.12% | 1.08% | 1.45% | 1.60% | 34.60% |

Table 7: Performances of test-time tuned model on MassSpecGym (Bushuiev et al., 2024) subsets obtained grouping the data depending on different features of the spectra. In particular, the performances have been analyzed depending on adduct, collision energy and instrument used to obtain each spectrum, and parent mass of the compound.

## C.2 DATA POINTS SELECTION DURING TEST-TIME TUNING

Figure 8 illustrates the evolution of the number of selected training points across test-time tuning iterations for NPLIB1 and MassSpecGym. In the case of NPLIB1, the curve rapidly approaches the size of the candidate pool ($\sim 15\,000$), indicating that most points are relevant for adaptation in an in-distribution setting. Conversely, for MassSpecGym, the selection grows more gradually and continues to increase throughout the iterations, reflecting the diversity and domain shift between training and test sets. This behavior highlights the adaptive nature of the selection process and its dependence on dataset characteristics.

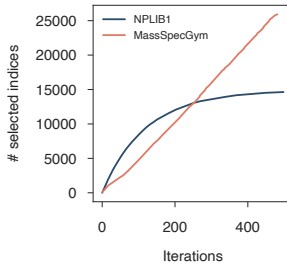

Figure 8: Length of the set of selected indices over the test-time tuning iterations, for NPLIB1-Full and MassSpecGym. In the first case, almost all the points in the candidate pool are selected, showing their relevance for the test points. In the case of MassSpecGym, instead, the method keeps selecting new points at every iteration, because of the larger size of the test set, but also implies diversity among the test points and between the test and train points.

## C.3 NPLIB1 FILTERED BY $H^+$ ADDUCT

We report in Table 8 the results of our models on the dataset NPLIB1-$H^+$, obtained as described in Appendix A.

| | NPLIB1-H$^+$ | | | |
|---|---|---|---|---|
| | Top-1 Accuracy ↑ | Top-5 Accuracy ↑ | Top-10 Accuracy ↑ | Valid SMILES ↑ |
| Fine-tuning | 14.11% | 27.56% | 32.89% | 89.24% |
| Test-time tuning | 14.61% | 27.19% | 32.55% | 89.96% |

Table 8: Performances of the fine-tuned and test-time tuned models (starting from the pre-trained model on the simulated dataset) on the experimental dataset NPLIB1-H$^+$, obtained by filtering the original NPLIB1-Full dataset depending on the adduct used to obtain each spectrum.

## D    ABLATIONS

To assess the impact of key hyperparameters on model performance, we conducted ablation studies varying batch size, number of K-means clusters, and number of epochs after which embeddings (and fingerprints) are updated. These experiments were performed on the full versions of the datasets, and the corresponding results are presented in Figure 9. Based on these findings, the hyperparameter values selected for batch size, number of K-means clusters, and update frequency in the test-time tuning runs described in the main text are summarized in Table 9. The best-performing parameters identified through these ablations were subsequently applied to the derived dataset variants (NPLIB1-H$^+$, NPLIB1-DiffMS, and MassSpecGym-DiffMS).

| | NPLIB1-Full | MassSpecGym | Extended NPLIB1-Full |
|---|---|---|---|
| batch size | 128 | 64 | 256 |
| # clusters | 1000 | 500 | 1000 |
| # epochs | 20 | 50 | 100 |

Table 9: Final hyperparameters used for test-time tuning experiments reported in the main text, depending on the dataset. Values were chosen according to the results in Figure 9.

### D.1    FINGERPRINT ALIGNMENT

Table 10 summarizes the performance of pre-trained and fine-tuned models on simulated dataset and NPLIB1 respectively, with and without the use of the loss term based on the fingerprints, described in Section B.1. Different values of the weight $\lambda$ (see Equation 1) are explored. The inclusion of the alignment term improves performances, in particular, when $\lambda = 0.1$ during pre-training, and even more when $\lambda$ is set to 1 when fine-tuning.

| | $\lambda$ | Top-1 Accuracy ↑ | Top-5 Accuracy ↑ | Top-10 Accuracy ↑ |
|---|---|---|---|---|
| | 0.0 | 34.05% | 57.73% | 62.69% |
| Pre-training simulations dataset | 0.1 | **36.02%** | **59.45%** | **64.36%** |
| | 1 | 34.46% | 57.40% | 61.98% |
| | 0.0 | 11.62% | 25.35% | 29.88% |
| Fine-tuning NPLIB1-FULL | 0.1 | 11.64% | 24.99% | 29.44% |
| | 1 | **12.42%** | **25.91%** | 30.83% |
| | 10 | 11.74% | 25.84% | **31.30%** |

Table 10: Performances of pre-training on simulated dataset (top section) and fine-tuning on NPLIB1 using different values of $\lambda$, weighting the loss term based on fingerprints prediction.

## E    COMPUTATIONAL REQUIREMENTS

All experiments were run on NVIDIA A100-SXM4-80GB GPUs. Test-time tuning requires approximately 13–15 hours on 4 GPUs for both NPLIB1 and MassSpecGym. Fine-tuning on MassSpec-Gym takes  8 hours on 4 GPUs, while on NPLIB1 it requires  6 hours on a single GPU, reflecting the smaller dataset size. The additional overhead of TTT compared to fine-tuning is due to the iterative retrieval and gradient update steps, and scales with the size of the candidate pool and number of clusters.

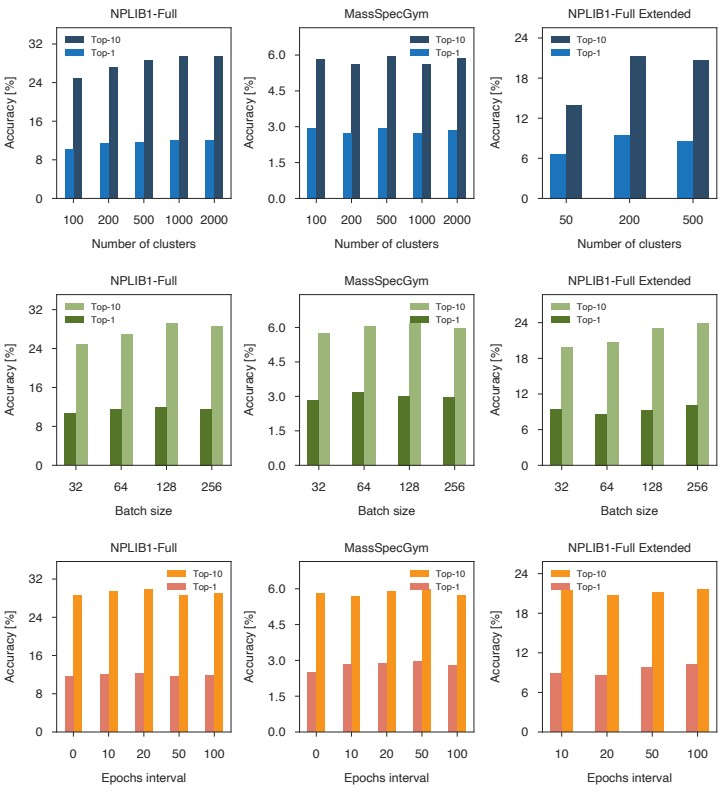

Figure 9: Performances of the test-time tuned models on NPLIB1-Full, MassSpecGym and extended NPLIB1-Full dataset, by varying number of clusters (iterations), batch size (number of training points selected per test point) and number of epochs after which the embeddings are updated.

