# OpenReview forum: "Test-Time Tuned Language Models Enable End-to-end De Novo Molecular Structure Generation from MS/MS Spectra"
_ICLR.cc/2026/Workshop/FM4Science — ICLR 2026 Workshop FM4Science Poster_

### Official Review · Reviewer_j73f · 2026-02-22
**Test-Time Tuned LMs for End-to-End De Novo Structure Generation from MS/MS**

**Rating:** 6
**Confidence:** 2

**Review:**

**Summary**

This paper proposes an end-to-end transformer encoder--decoder that generates SMILES directly from an input MS/MS spectrum and its molecular formula, with an auxiliary molecular fingerprint head. To address domain shift between training and deployment spectra, it introduces a test-time tuning (TTT) strategy: for each unlabeled test spectrum, the model predicts fingerprint logits, uses them to select the most relevant samples from a candidate pool (typically the training set), and performs a small number of gradient updates before generating candidates.

**Strengths**

1. The problem setting is practically important: MS/MS structure elucidation is hard and domain shift is common.
2. The approach is conceptually simple and engineering-friendly: end-to-end generation plus TTT-based adaptation.
3. The paper clearly illustrates when TTT helps: in-distribution (NPLIB1) TTT is comparable to standard fine-tuning, while under strong domain shift (MassSpecGym) fine-tuning can degrade but TTT improves performance.
4. Evaluation goes beyond Top-k accuracy by reporting structural similarity proxies (e.g., Tanimoto / MCES), arguing that generated candidates remain chemically informative even when the exact structure is not recovered.

**Weaknesses**
1. Deployment cost and scalability: TTT requires retrieval over a potentially large candidate pool and gradient updates at inference time; more explicit wall-clock accounting and scaling trends would strengthen the claim of practical applicability.
2. Dependence on informative neighbors: the benefit of TTT likely relies on having structurally relevant spectra in the candidate pool; performance may be constrained when such neighbors are absent.
3. Robustness of the selection step: since selection is driven by predicted fingerprints (which may be unreliable under OOD), additional diagnostics on selection quality and sensitivity to clustering/similarity choices would be helpful.

---

### Official Review · Reviewer_yzpH · 2026-02-24
**This paper is not bad**

**Rating:** 6
**Confidence:** 2

**Review:**

Actually, I'm not familiar with MS Spectra or Molecular Structure Generation.

From the perspective of writing, this paper is well organized and with enough quality for the workshop.

If there are any other reviews that are more professional, plz ingore this one.

---

### Official Review · Reviewer_Ng4E · 2026-02-24
**Promising Test-Time Adaptation Framework with Reproducibility Considerations**

**Rating:** 7
**Confidence:** 3

**Review:**

## Summary

This paper studies robustness under distribution shift for transformer-based sequence generation and proposes a transductive test-time tuning (TTT) procedure. Instead of relying solely on static pre-training or fine-tuning, the method retrieves informative support examples using model-predicted fingerprints and performs iterative adaptation at inference time. Empirically, the authors report consistent improvements of TTT over PT/FT baselines across several dataset variants, with larger gains in more out-of-distribution settings. The overall contribution is positioned as a practical instantiation of retrieval-guided adaptation for structured generation tasks.

## Main Review

The problem being addressed — robustness under distribution shift — is central to modern machine learning, and the focus on test-time adaptation rather than further scaling or retraining is well motivated. The PT/FT/TTT decomposition is clearly presented, and the adaptation loop (retrieve, adapt, iterate) is intuitive and easy to follow. The reported empirical pattern, where TTT consistently improves over static baselines and shows larger gains under stronger shifts, is encouraging.

There are several positive aspects of the work:
	•	The idea of retrieval-guided adaptation using predicted fingerprints is conceptually sensible, instead of adapting blindly at inference time.
	•	The empirical gains appear directionally consistent across settings rather than isolated to a single benchmark
	•	The appendix includes ablations of TTT hyperparameters and adaptation behavior, which is helpful
	•	The paper is generally clearly written and the methodological flow is easy to understand.

However, there are several important shortcomings that should be addressed.

The novelty relative to prior TTT and transductive adaptation work appears somewhat incremental. The paper would benefit from a sharper articulation of what is algorithmically new here beyond applying known test-time adaptation ideas to this specific structured generation setting. As written, it is not entirely clear how this approach fundamentally differs from existing methodology.

The iterative adaptation at inference time can be computationally expensive, yet there is no explicit analysis of latency, memory usage, or accuracy–vs.–adaptation-budget trade-offs. From a practical standpoint, it is not clear how costly the TTT loop is relative to its gains, and this affects deployability.

Finally, the results are distributed across multiple dataset variants with slightly different assumptions. A consolidated comparison table with identical protocol settings would make the high-level conclusions easier to interpret.

---

### Official Review · Reviewer_ZTPM · 2026-02-25
**Review of "Test-Time Tuned Language Models Enable End-to-end De Novo Molecular Structure Generation from MS/MS Spectra"**

**Rating:** 5
**Confidence:** 4

**Review:**

This paper proposes an end-to-end transformer framework for **de novo small-molecule structure generation from MS/MS spectra**, taking as input a tandem mass spectrum plus the (assumed-known) **molecular formula**, and directly generating **SMILES** (with additional fingerprint supervision). To handle domain shift between simulated and experimental spectra (and between training and deployment conditions), the paper introduces a **test-time tuning (TTT)** strategy that adapts the model transductively by selecting informative samples from a candidate pool using a predicted-fingerprint feature space. Empirically, the method reports large relative improvements over prior baselines on challenging OOD benchmarks (MassSpecGym) while also achieving strong chemical similarity (Tanimoto/MCES) even when exact Top–1 structure recovery is rare.

### Summary of method and claims
The model is a transformer encoder–decoder trained to map a tokenized spectrum (peaks as text) plus chemical formula to a SMILES sequence. The encoder additionally feeds an MLP head that predicts molecular fingerprints; training uses a SMILES cross-entropy loss plus a fingerprint BCE term:

$$
L = L_{\\mathrm{CE,SMILES}} + \\lambda L_{\\mathrm{BCE,Fingerprints}}
$$

Key components beyond the base seq2seq model:
- **Simulation-based pre-training**: pre-train on large-scale simulated MS/MS spectra (millions of spectra; hundreds of thousands of unique SMILES), then adapt to experimental datasets.
- **Formula-constrained decoding**: during generation, restrict next-token choices to enforce consistency with the provided chemical formula.
- **Rejection sampling**: generate many candidates and keep the first valid ones (e.g., retain up to Top–10 valid SMILES).
- **Test-time tuning (transductive adaptation)**: for each test set, predict fingerprint logits for unlabeled test spectra, cluster test points (K-means) in this fingerprint-logit space, and iteratively retrieve/select training samples from a candidate pool via cosine similarity; perform gradient updates using the selected labeled samples and repeat over clusters.

Evaluation is performed on NPLIB1 and MassSpecGym (and DiffMS-filtered variants), reporting Top–$k$ accuracy plus structural similarity metrics (Tanimoto similarity and MCES distance) and validity rates.

### Strengths
- **End-to-end formulation with practical constraints**: directly generating SMILES from spectra (+ formula) removes reliance on fragment annotation pipelines and intermediate fingerprint-only systems, while formula-constrained decoding is a strong, domain-appropriate inductive bias.
- **Clear treatment of distribution shift**: the paper explicitly distinguishes in-distribution (NPLIB1 holdout) vs out-of-distribution (MassSpecGym split by MCES distance) regimes and shows that naive fine-tuning can hurt under shift, motivating adaptation.
- **Transductive test-time tuning is well-motivated here**: unlike many ML tasks, MS/MS test inputs are available at inference time and unlabeled; using them to steer which training data to adapt on is a reasonable approach for realistic “unknown molecule” scenarios.
- **Chemical usefulness beyond Top–1**: the emphasis on Tanimoto/MCES and meaningful/close-match rates is appropriate given the intrinsic ambiguity of structure elucidation; the candidates can still narrow expert search even when exact recovery is unlikely.
- **Reproducibility signals**: the paper provides substantial dataset/version details, multiple dataset variants (including DiffMS-style filtering), and reports additional analyses (adduct/instrument/mass/energy subsets; ablations).

### Weaknesses and concerns (major)
- **Potential data leakage / benchmark contamination (critical to rule out)**: the paper’s strongest gains rely on retrieval from a **candidate pool** (and, in some settings, expanding that pool by merging datasets) plus large-scale **simulation-based pre-training**. This combination creates a real risk that some reported improvements are inflated by *exact or near-exact molecule overlap* between (i) the candidate pool used for test-time tuning, (ii) the simulated pre-training corpus (including public forward simulators / weights such as ICEBERG/CFM-ID variants), and (iii) the benchmark test sets. The reviewer did an overlap check (by exact InChIKey match, without additionally enforcing an MCES-style distance constraint) between the paper’s simulated pre-training corpus (“Multimodal simulated spectra” from Alberts et al., 2024b) and downstream benchmark test structures (e.g., CANOPUS and MassSpecGym), and also between a MassSpecGym train split used as a tuning candidate pool and CANOPUS test structures; the observed overlap rates can be non-trivial (e.g., about 15% of CANOPUS test and about 10% of MassSpecGym test present in the simulated corpus; and 79.83% of CANOPUS test present in MassSpecGym train in that comparison setting). Even if some experiments state “no overlap with the target test set,” the paper should **explicitly quantify and publish** molecule-level overlap statistics and rerun key results under strict deduplication rules:
  - remove any candidate-pool molecules whose identifiers (e.g., InChIKey) match the target test molecules,
  - report performance with and without any simulated-pretraining molecules that overlap evaluation test sets,
  - clarify whether forward simulators (and their training data) may have seen benchmark test molecules, and if so, provide a leak-free variant for fair comparison.
- **Transductive setting needs clearer “allowed information” boundaries**: TTT adapts using unlabeled test inputs plus labeled candidate-pool samples. This is valid in some deployment scenarios, but it is not directly comparable to purely inductive methods unless the evaluation protocol explicitly allows transductive adaptation. The paper should clearly define: (i) whether all test inputs are available upfront, (ii) whether adaptation is per-test-point or per-test-set, and (iii) what compute/time budget is assumed.
- **Dependence on molecular formula is an incomplete assumption**: the method assumes the correct chemical formula is provided. In many realistic MS/MS workflows, the formula is itself a *prediction* with uncertainty (adduct ambiguity, isotopes, missing peaks). Beyond synthetic “formula noise” ablations, it would strengthen the paper to **report end-to-end performance when the formula is produced by off-the-shelf predictors** (e.g., `SIRIUS`, `MIST-CF`) and to quantify how formula error propagates to Top-$k$ and similarity metrics.
- **Attribution of gains across multiple “helper” mechanisms is under-identified**: the full system includes simulation pre-training, fingerprint alignment, formula constraints, rejection sampling, and TTT retrieval/updates. While some ablations exist, the core benchmark gains (especially on MassSpecGym OOD) would be more convincing with a clean component-by-component attribution on the same evaluation protocol.
- **Compute and scalability of TTT is a real concern**: TTT involves clustering test fingerprints and repeated retrieval from a potentially huge candidate pool, plus gradient updates and periodic embedding refresh. The paper acknowledges this but does not provide enough wall-clock / GPU-hours per dataset (and how it scales with pool size and number of clusters) to assess practicality.

### Weaknesses (minor)
- **Fairness of baseline comparisons**: Table-based comparisons mix numbers from prior work, DiffMS-implemented baselines, and multiple dataset variants. Even with best effort, subtle preprocessing differences (canonicalization, formula availability, spectrum merging across collision energies) can change Top–$k$ accuracy materially; the paper should emphasize “apples-to-apples” comparisons on the exact same splits/filters where possible.
- **Metric choice could be expanded**: Top–$k$ accuracy is stringent; Tanimoto/MCES are helpful. It would also be useful to report scaffold-level correctness, functional-group recovery, or candidate-set diversity/coverage to avoid overly redundant candidate lists.

### Suggestions for improvement
- **Clarify and standardize the evaluation protocol for transduction**:
  - explicitly state whether the model sees the full unlabeled test set before predicting any item,
  - report results in two modes: (i) inductive (no TTT) and (ii) transductive (TTT), with clear compute budgets.
- **Stronger ablations / controls** (same datasets, same preprocessing):
  - PT vs PT+formula-constraint vs +rejection sampling vs +fingerprint alignment vs +TTT,
  - TTT with predicted fingerprints vs oracle fingerprints (upper bound) to isolate selection quality,
  - selection based on spectrum similarity (spec2vec/MS2DeepScore) vs fingerprint-logit similarity.
- **Robustness to imperfect formula**:
  - evaluate with noisy/partial formulas, and explicitly with formulas predicted by off-the-shelf tools such as `SIRIUS` and `MIST-CF` (or other MS1/MS2 formula predictors),
  - analyze performance when adduct type is unknown (given the large Na+ drop reported).
- **Practicality reporting**:
  - include retrieval/indexing details (ANN library, embedding refresh frequency),
  - provide runtime per molecule (and per dataset) and memory requirements as candidate pool grows.
- **Candidate quality analysis**:
  - report diversity metrics for the Top–10 set (to ensure candidates are not near-duplicates),
  - add qualitative case studies where Top–1 fails but Top–10 provides actionable substructures.

### Summary - Pros and cons

**Pros**
- **Strong end-to-end framing** for MS/MS→SMILES that reduces reliance on multi-stage heuristics.
- **Well-motivated test-time tuning** approach for handling severe OOD shift (MassSpecGym).
- **Useful similarity-based evaluation** (Tanimoto/MCES) that reflects real-world “helpfulness” beyond exact matches.
- **Detailed dataset/version handling** and several supporting analyses/ablations.

**Cons**
- **Comparability/fairness questions** due to transductive adaptation and mixed baseline protocols.
- **Heavy reliance on correct molecular formula**, with unclear robustness to realistic formula uncertainty.
- **System complexity and compute overhead** of TTT + retrieval is not fully quantified.
- **Need stronger attribution** of where gains come from (constraints vs pretraining vs TTT vs sampling).

---

### Official Review · Reviewer_UjQg · 2026-02-25
**This paper talks about a new end-to-end transformer-based framework for generating molecular structures directly from Tandem Mass Spectrometry (MS/MS) spectra and molecular formulae. To overcome the large domain shift between simulated training data and experimental data, the authors here propose a test-time tuning strategy. This strategy adapts the model to unlabeled test points by selecting informative samples from a training candidate pool. The model achieves good results on the MassSpecGym and NPLIB1 benchmarks, demonstrating that adaptive transductive learning significantly outperforms traditional fine-tuning in out-of-distribution (OOD) scenarios.**

**Rating:** 6
**Confidence:** 4

**Review:**

The paper addresses a critical bottleneck in chemistry of the de novo identification of unknown small molecules from MS/MS spectra. While data-driven methods have made an impact here, most existing methods here rely on complex pipelines or predicted fingerprints. This work shows an end-to-end generative approach that leverages the adaptability of foundation model-like architectures. The introduction of test-time tuning to this domain is particularly interesting, as it provides a nice way to handle the domain shift that often occurs in experimental science fields.
The authors have fairly rigorous benchmarking, ablation studies and relevant methodological detail.
The primary originality lies in the application of transductive learning to small-molecule structure generation. While TTT has seen success in computer vision and proteomics, its adaptation here using predicted molecular fingerprints to identify informative training neighbors is a nice solution to the lack of labels when at inference time
Pros: End-to-end architecture without manual fragment annotation; robust to domain shift via TTT bridging simulated and experimental spectra; chemically consistent generation through formula constraints and fingerprint alignment; and strong relative gains in structural similarity metrics (Tanimoto, MCES), offering meaningful structural hints even without exact matches.
Cons: Retrieval-based TTT raises a lot of scalability concerns as the candidate pool grows; low absolute Top-1 accuracy (e.g., 3.16%) reflects task difficulty and limits standalone usability; and performance depends heavily on the presence of structurally relevant neighbors in the candidate set. The language is relatively complex for one not fully familiar with the domain, even in the abstract.

---

### Official Review · Reviewer_Jdyu · 2026-02-25
**Good quality paper of mitigating Sim-to-Real Gap in Metabolomics**

**Rating:** 7
**Confidence:** 3

**Review:**

Summary
This paper successfully applies Test-Time Tuning (TTT) to the problem of de novo molecular structure generation from MS/MS spectra. By dynamically adapting the model to the test distribution using retrieved candidates, the authors effectively address the domain shift between simulated training data and real experimental spectra.

Strengths
1. TTT significantly outperforms static fine-tuning, achieving a 41% relative gain on NPLIB1 when using an extended candidate pool.
2. While exact matches are rare, the generated structures show high Tanimoto similarity to ground truth (83% improvement over SOTA), making them practically useful for chemists narrowing down candidates.
3. The end-to-end transformer architecture removes the need for complex intermediate steps like fingerprint prediction or fragment annotation.


Weaknesses
1. Top-1 accuracy remains low (~3% on MassSpecGym), indicating that while the method is a step forward, fully automated structure elucidation remains unsolved.
2. The retrieval and adaptation steps in TTT introduce computational overhead compared to standard inference, potentially limiting high-throughput scalability

---

### Meta-Review · Area_Chair_ZXVm · 2026-02-28

**Recommendation:** Accept (Poster)
**Confidence:** 3

**Metareview:**

This submission proposes an end-to-end transformer encoder–decoder that generates SMILES directly from MS/MS spectra and a provided molecular formula, and introduces a retrieval-guided test-time tuning (TTT) procedure to mitigate sim-to-real / out-of-distribution shift. Reviewers agree the problem is important and the pipeline is appealingly direct (avoiding multi-stage fingerprint/annotation pipelines), and multiple reviews find the empirical pattern convincing: TTT improves over static fine-tuning particularly under strong distribution shift, with meaningful gains in similarity metrics even when exact Top-1 recovery remains low.

The main concern comes from one detailed review arguing that the strongest reported gains may be confounded by molecule overlap / benchmark contamination due to the combination of (i) large simulated pretraining corpora and (ii) retrieval from candidate pools used during transductive adaptation; the reviewer requests explicit molecule-level overlap statistics and strict deduplication controls. Additional reviewers also note practical issues: TTT compute/scalability and clearer definition of the allowed transductive protocol/budget, plus the strong assumption that the correct molecular formula is known.

---

### Decision · Program_Chairs · 2026-03-03

Accept (Poster)